# Towards Mitigating Audio Adversarial Perturbations

**Zhuolin Yang**
Shanghai Jiao Tong University

**Bo Li**
UC, Berkeley

**Pin-Yu Chen**
IBM Research

**Dawn Song**
UC, Berkeley

## Abstract

Audio adversarial examples targeting automatic speech recognition systems have recently been made possible in different tasks, such as speech-to-text translation and speech classification. Here we aim to explore the robustness of these audio adversarial examples generated via two attack strategies by applying different signal processing methods to recover the original audio sequence. In addition, we also show that by inspecting the temporal consistency in speech signals, we can potentially identify non-adaptive audio adversarial examples considered in our experiments with a promising success rate.

## 1 Introduction

Deep Neural Networks (DNNs) have been widely adopted in a variety of applications (Krizhevsky et al., 2012; Hinton et al., 2012; Levine et al., 2016). However, recent work has demonstrated that DNNs are vulnerable to adversarial perturbations (Szegedy et al., 2014; Goodfellow et al., 2015). An adversary can add negligible perturbations to inputs and generate adversarial examples to mislead DNNs, especially in image-based machine learning tasks (Goodfellow et al., 2015; Carlini & Wagner, 2017; Liu et al., 2017; Chen et al., 2017b;a)

Beyond images, given the wide application of DNN-based audio recognition systems, such as Google Home and Amazon Alexa, audio adversarial examples have also been studied recently (Carlini & Wagner, 2018; Alzantot et al., 2018; Cisse et al., 2017). In this paper, we aim to explore the robustness of such audio adversarial examples against different signal sampling and recovery methods. As a first attempt towards mitigating audio adversarial examples, we conduct extensive experiments to show that with different signal processing techniques, such as amplitude quantization, local smoothing, and down-sampling, the adversarial examples considered in our experiments can be recovered with high-quality recognition results without affecting the benign examples too much. We focus on non-adaptive attacks, and we consider mitigating adaptive attacks as our future work. In addition, we utilize the sequential dependency in audio data to discriminate audio adversarial examples and find that it is possible to identify audio adversarial examples based on temporal consistency checking. We perform the temporal consistency verification on both the LIBRIS (Graetz et al., 1986) and Mozilla Common Voice datasets against two state-of-the-art attack methods (Carlini & Wagner, 2017; Alzantot et al., 2018) considered in our experiments and show that such an approach achieves promising identification of non-adaptive attacks.

## 2 Background on Audio Adversarial Examples

Generally speaking, an adversarial example for a neural network is an input $x^a$ that is similar to a natural input $x$ but will yield different output after passing through the neural network. Currently, there are two different types of audio adversarial attacks: the Speech-to-label attack and the Speech-to-text attack, and we consider the targeted attack for both cases. The Speech-to-label attack aims to find an adversarial example $x^a$ close to the original audio $x$ but yields a different (wrong) label. Alzantot et al. (2018) proposed a genetic algorithm to generate such adversarial examples. The Speech-to-text attack requires the transcribed output of the adversarial audio to be the same as the desired output, which has been made possible by Carlini & Wagner (2018) using optimization-based techniques.

## 3    AUDIO ADVERSARIAL EXAMPLES UNDER SIMPLE TRANSFORMATIONS

As a first attempt, we applied some primitive signal processing transformations on audio data to recover them from adversarial perturbations without decreasing their quality too much. These transformations are useful, easy to implement, fast to operate and have delivered some interesting findings.

- **Quantization:** By rounding the amplitude of audio sampled data into the nearest integer multiple of $q$, the adversarial perturbation can be disrupted since its amplitude is usually small in the input space. We choose $q = 128, 256, 512, 1024$ as our parameters.

- **Local smoothing:** We use a sliding window of a fixed length for local smoothing to reduce the adversarial perturbation. For an audio sample $x_i$, we consider the $k-1$ samples before and after it, denoted by $x_{i-k+1}, \ldots, x_i, \ldots, x_{i+k-1}$, as a local reference sequence and replace $x_i$ by the smoothed value (average, median, etc) of its reference sequence.

- **Down sampling:** Based on sampling theory, it is possible to down-sample an audio file without sacrificing the quality of the recovered signal while mitigating the adversarial perturbations in the reconstruction process. In our experiments, we down-sample the original 16kHz audio data to 8kHz and then perform signal recovery.

## 4    TEMPORAL CONSISTENCY OF AUDIO ADVERSARIAL EXAMPLES

Consider how an adversarial audio is generated: an adversary adds some unnoticeable perturbation to the original audio to ensure it to be transcribed to the adversarial target. In another aspect, we hypothesize that adversarial audio can be fragile and it needs complete audio information to resolve temporal dependency. Under this hypothesis, if we cut the audio data into two sections, the natural audio data is expected to be transcribed normally in each section except for some phrases near the cut position, while a section of adversarial audio is expected to be transcribed more differently.

Consequently, we propose a temporal-consistency-based approach to characterize properties of audio adversarial examples. Let $x$ be the input audio and $x^*$ be the first half of $x$. Let $f(\cdot)$ denote the speech recognizing and transcribing process. To measure the similarity between $f(x)$ and $f(x^*)$ (they usually have different lengths), we only compare the prefix $f^*(x)$ of $f(x)$ that has the same length as $f(x^*)$ and ignore the rest. We use word error rate (WER), character error rate (CER), and longest common prefix ratio (LCP ratio) to evaluate the differences between $f^*(x)$ and $f(x^*)$. We will show that all of these metrics are useful for differentiating audio adversarial examples and achieve high AUC scores for identifying non-adaptive attacks considered in our experiments.

## 5    EXPERIMENTAL RESULTS AND DISCUSSION

### 5.1    ANALYSIS FOR TRANSFORMATIONS ON ADVERSARIAL AUDIO

In our experiments, we measure the robustness of audio adversarial attacks by Carlini & Wagner (2018) and Alzantot et al. (2018). For Carlini & Wagner's attack, we separately choose 50 audio files from two audio datasets (Common Voice, LIBRIS) and generate attacks based on CTC-loss.

Table 1: Evalation on Common Voice. The ratio between the transcribing error rate referenced by ground truth and the adversarial target is shown in brackets.

| Mitigation Methods | OriginWER(%) | OriginCER(%) | AdvWER(%) | AdvCER(%) |
|---|---|---|---|---|
| Without mitigation | 37.7 (0.18) | 18.5 (0.10) | 95.8 (10.1) | 83.0 (31.6) |
| Median-4 | 43.4 (0.21) | 20.4 (0.11) | 83.0 (0.46) | 46.5 (0.34) |
| Down sampling | 47.2 (0.23) | 23.3 (0.13) | 77.6 (0.43) | 43.9 (0.31) |
| Quantization-128 | 47.3 (0.24) | 25.7 (0.15) | 80.7 (0.52) | 49.0 (0.40) |
| Quantization-256 | 52.5 (0.28) | 29.2 (0.18) | 73.4 (0.43) | 43.6 (0.31) |
| Quantization-512 | 64.1 (0.35) | 37.5 (0.24) | 73.7 (0.43) | 44.2 (0.31) |
| Quantization-1024 | 72.1 (0.42) | 50.4 (0.37) | 76.9 (0.45) | 53.0 (0.41) |

Table 2: Examples of the proposed attack mitigation approaches

| Type | Decode results |
|------|----------------|
| Adversarial | enough said the boy |
| Original | they snuffed about the fur tree and rustled among the branches |
| Mitigated | they snuffed about the vertre and ruslled amon the branches |
| | |
| Original | this because you were not born |
| the first half of Original | this because you were |
| Adversarial | enough said the boy |
| First half of Adversarial | yens egasia or |

Table 3: AUC Results of Temporal Consistency Method

| Dataset | LSTM | TC(WER) | TC(CER) | TC(LCP ratio) |
|---------|------|---------|---------|---------------|
| Common Voice | 0.712 | **0.936** | 0.916 | 0.859 |
| LIBRIS | 0.645 | **0.911** | 0.902 | 0.729 |

We evaluate several signal processing methods based on WER and CER metrics and also report the ratio between the transcribing error rate referenced by ground truth and the adversarial target (shown in brackets). The results are shown in Tables 1 and 4. From the results, we can see that most of our attack mitigation methods (e.g., Median-4, Down sampling and Quantization-256) can effectively reduce the adversarial perturbation without affecting the original audio's transcribed results too much. Table 2 shows some adversarial examples recovered by Quantization-256. The results are reasonable and appear non-adversarial, since they are merely some mistakes caused by similar phrase replacement or spelling errors. In order to further fix the transcribed results caused by these mistakes, we used DeepSpeech's language model. The results are mentioned in Tables 6 and 7.

In Alzantot et al.'s attack, we implemented their attack with 500 iterations and limit the magnitude of adversarial perturbation within 5 (smaller than the quantization we use as a mitigation), then we generated 50 adversarial examples on each attack task as shown in Figures 1 and 2. The attack success rate is $84\%$ on the labeled audio datasets, and the average unchanged label rate is $10.6\%$. For illustration, we use Quantization-256 as our mitigation method. The results against adversarial examples considered within our experiments are very promising, as can be observed in Figures 3 and 4. The attack success rates decreased to only $2.1\%$, and $63.8\%$ of the adversarial audio files are converted back to their original label. We also measure the possible effects on original audio files caused by our mitigation methods: the original audio without mitigation can be classified correctly with a rate of $89.2\%$, and the rate decreased to $89.0\%$ after applying our mitigation methods, which means the effects of our mitigation on benign instances are negligible.

## 5.2 ANALYSIS FOR TEMPORAL CONSISTENCY ON ADVERSARIAL AUDIO

Our temporal consistency analysis is based on Carlini's CTC-loss attack and used 100 randomly and separately picked audio files from Common Voice and LIBRIS audio datasets. Some identified samples are shown in Table 2. We find that while transcribing the sentence, the first half of a natural audio has a slight change when compared to the ground truth, but for the first part of an adversarial audio, the change is significant. Also, this method can be effective even if the adversarial audio hides speech from being transcribed (e.g., Carlini & Wagner's silence attack (2018)): this method still reveals the ground truth.

In our temporal consistency comparison, we used three metrics: WER, CER, and LCP ratio. Tables 9 and 10 show their differences between natural and adversarial audio files. We then use these three metrics to evaluate our method's performance by using AUC scores, together with a trained baseline LSTM which has 64 hidden layer features as our baseline model. The comparison is presented in Table 3. We find that the single LSTM model's performance is not ideal, whereas using the WER metric achieved 0.936 on Common Voice and 0.911 on LIBRIS. The results suggest a simple but promising method for characterizing adversarial audio attacks.

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

# 6 APPENDIX

Table 4: Evalution on LIBRIS

| Mitigation Methods | OriginWER(%) | OriginCER(%) | AdvWER(%) | AdvCER(%) |
|---|---|---|---|---|
| Without mitigation | 12.4 (0.07) | 7.05 (0.06) | 105.3 (65.8) | 91.7 (611.3) |
| Median-4 | 16.4 (0.09) | 8.0 (0.07) | 57.9 (0.34) | 27.5 (0.26) |
| Downsample | 24.2 (0.13) | 13.0 (0.11) | 60.9 (0.35) | 31.2 (0.28) |
| Quantization-128 | 13.4 (0.07) | 7.6 (0.06) | 66.1 (0.40) | 37.1 (0.38) |
| Quantization-256 | **16.3 (0.09)** | **8.9 (0.08)** | **48.6 (0.28)** | **24.0 (0.22)** |
| Quantization-512 | 27.5 (0.15) | 13.8 (0.12) | 47.0 (0.27) | 23.0 (0.19) |
| Quantization-1024 | 46.8 (0.27) | 25.4 (0.23) | 52.3 (0.30) | 30.0 (0.28) |

Table 5: Mitigated Samples

| Type | Mitigation results |
|---|---|
| Adv. | enough said the boy |
| | |
| Original | they snuffed about the fur tree and rustled among the branches |
| Mitigated | they snuffed about the vertre and ruslled amon the branches |
| | |
| Original | and he leaned against the wa lost in reveriey |
| Mitigated | and he leaned against the wall losting reavelete |
| | |
| Original | and then what happens then |
| Mitigated | as the what happenis them |

Table 6: Evaluation on Common Voice with language model

| Mitigation Methods | OriginWER(%) | OriginCER(%) | AdvWER(%) | AdvCER(%) |
|---|---|---|---|---|
| Without mitigation | 27.5 (0.14) | 14.3 (0.08) | 95.9 (9.13) | 80.1 (9.65) |
| Median-4 | 27.0 (0.14) | 14.6 (0.08) | 73.6 (0.59) | 42.4 (0.29) |
| Downsample | 31.2 (0.17) | 17.6 (0.10) | 69.6 (0.53) | 41.2 (0.29) |
| Quant-128 | 34.4 (0.19) | 21.3 (0.12) | 75.9 (0.70) | 45.3 (0.36) |
| Quant-256 | 42.9 (0.25) | 26.7 (0.16) | 70.7 (0.57) | 41.8 (0.29) |
| Quant-512 | 52.4 (0.29) | 37.1 (0.24) | 68.5 (0.40) | 45.0 (0.32) |
| Quant-1024 | 62.4 (0.36) | 47.2 (0.35) | 70 (0.41) | 51.2 (0.39) |

Table 7: Evaluation on LIBRIS with language model

| Mitigation Methods | OriginWER(%) | OriginCER(%) | AdvWER(%) | AdvCER(%) |
|---|---|---|---|---|
| Without mitigation | 3.05 (0.02) | 1.46 (0.01) | 102.8 (15.1) | 86.5 (16.3) |
| Median-4 | 3.6 (0.02) | 1.7 (0.01) | 35.1 (0.23) | 19.0 (0.17) |
| Downsample | 11.8 (0.06) | 5.7 (0.05) | 41.2 (0.28) | 21.8 (0.19) |
| Quant-128 | 3.2 (0.02) | 1.5 (0.01) | 49.7 (0.35) | 28.2 (0.22) |
| Quant-256 | **3.5 (0.02)** | **1.7 (0.01)** | **29.1 (0.18)** | **15.4 (0.13)** |
| Quant-512 | 12.0 (0.07) | 6.6 (0.06) | 25.1 (0.15) | 13.3 (0.12) |
| Quant-1024 | 30.7 (0.19) | 20.3 (0.18) | 36.6 (0.23) | 24.1 (0.22) |

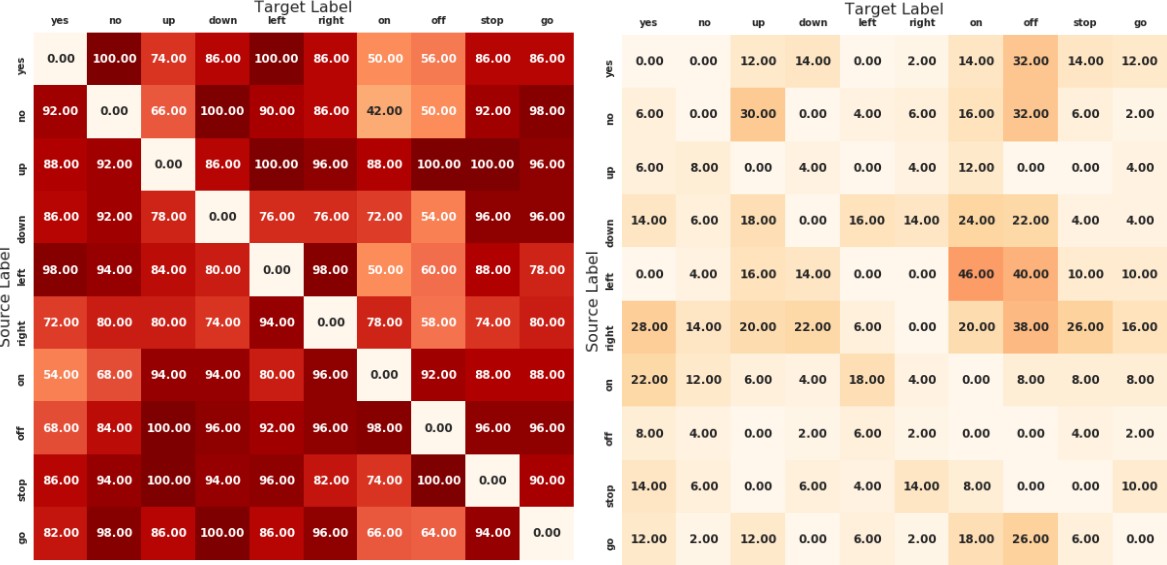

Figure 1: Successful attack rates          Figure 2: Unchanged label rates

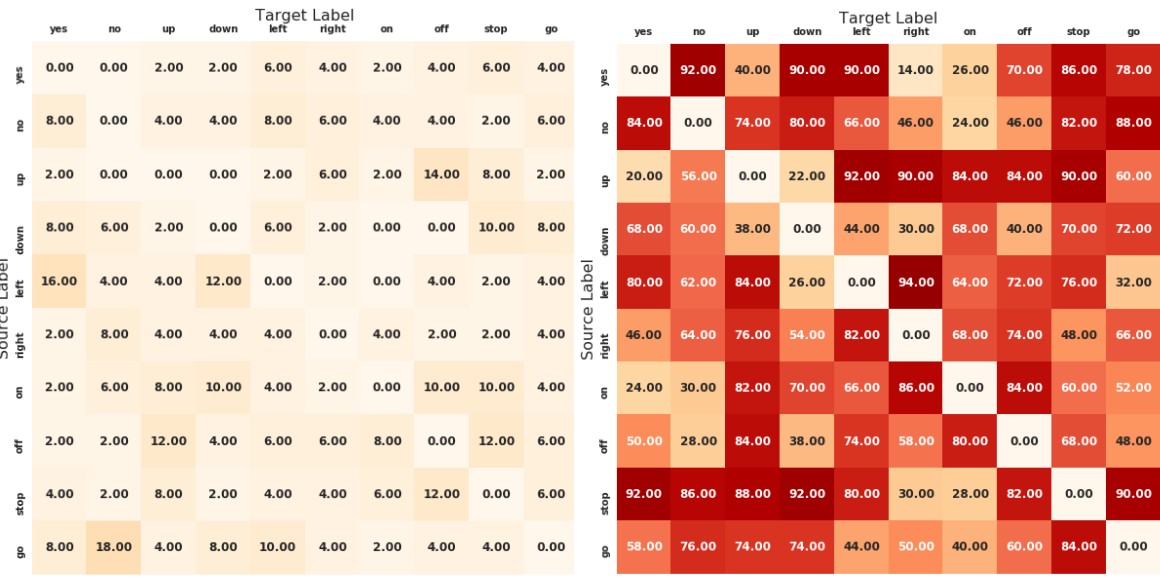

Figure 3: Successful attack rates after mitigation          Figure 4: Unchanged label rates after mitigation

Table 8: Identified samples

| Type | Decode results |
|------|----------------|
| Original | and the whole night the tree stood still and in deep thought |
| the first half of Original | and the whole night the trees stoo |
| Adv. | this is an adversarial example |
| the first half of Adv. | tho lits an aters o |
| | |
| Original | have you not met them anywae |
| the first half of Original | have you not |
| Adv. | *sil* |
| the first half of Adv. | have you not |
| | |
| Original | this because you were not born |
| the first half of Original | this because you were |
| Adv. | enough said the boy |
| the first half of Adv. | yens egasia or |

Table 9: Identified results on Common Voice

| Metrics | Original | Adversarial |
|---------|----------|-------------|
| WER(%) | 37.4 | 96.0 |
| CER(%) | 16.2 | 61.1 |
| LCP ratio(%) | 27.3 | 2.08 |

Table 10: Identified results on LIBRIS

| Metrics | Original | Adversarial |
|---------|----------|-------------|
| WER(%) | 21.6 | 65.8 |
| CER(%) | 5.02 | 33.4 |
| LCP ratio(%) | 38.5 | 20.3 |

