# OpenReview forum: "Towards Mitigating Audio Adversarial Perturbations"
_ICLR.cc/2018/Workshop — Reject_

### Official Review · AnonReviewer2 · 2018-03-06
**A simple but promising method for characterizing adversarial audio attacks**

**Rating:** 7
**Confidence:** 4

**Review:**

Overall I agree with the authors that their results suggest a simple but promising method for characterizing adversarial audio attacks.  I am not very well versed in this field, but the authors do seem to be using the state of the art attacks for their evaluation, Carlini and Wagner, and Alzantot, Balaji, and Srivastava and they are proposing a simple solution that can ward against these attacks.

This is a simple method that does seem to work.  It is good to know these attacks can be thwarted.  Were other methods tried that were not good (e.g. adding white noise)?

The impact on accuracy is not negligible, however, what I think is a great contribution for this paper is the fact that it is likely to be able to tell that you are being attacked by being able to do a difference on the mitigated and unmitigated transcriptions.  An interesting experiment here would have been to detect how many times the authors could detect adversarial attacks using this method, using a comparison in WER or CER.  Knowing that you are being attacked is important.  Out of scope, but an interesting application of your work.

The authors test their method against many standard data sets and report extensive results.  These results are worth publishing, so I am voting to accept.

---

### Official Review · AnonReviewer3 · 2018-03-10
**Interesting task but reviewer has concerns about experimental evaluations**

**Rating:** 4
**Confidence:** 3

**Review:**

The paper considers techniques to mitigate attacks on automatic speech recognition (ASR) systems which might allow an attacker to cause an ASR system to produce a transcript which doesn’t match what a human listener would select if she were presented with the same audio.
To the best of my knowledge, the task hasn’t received much attention in the literature, and this is an interesting task. However, there are a number of concerns that I have with this paper which cause me to suggest that the paper be rejected in its present form.
The techniques consider in the paper are fairly simple, but my main concerns lie in the experimental evaluation which in my opinion isn’t very thorough. E.g., Table 1 lists the results for 50 utterances, if I understand correctly, which is a very small sample.
Additionally, the evaluation procedure was hard to understand: E.g. in Table 1, what is the “OriginWER” vs. the “AdversarialWER”? Ideally, it would be interesting to examine the impact that the techniques proposed have on the recognition of non-adversarial samples, as well as on adversarial attacks. I would have liked a more detailed evaluation of both of these on larger sets.
Finally, in Table 1, the authors state that the procedure allows them to “reduce adversarial perturbation without affecting the original audio’s transcribed results too much”. However, when examining the results, it appears that there is a significant degradation in performance: Quantization 256, for example, results in a ~40% increase in WER, which is a significant degradation.

---

### Official Review · AnonReviewer1 · 2018-03-11
**unclear and too long**

**Rating:** 4
**Confidence:** 4

**Review:**

The paper presents "mitigation strategies" for adversarial attacks on speech-to-text systems.

The main idea seems to be to pass the (potentially adversarially perturbed) speech signal through a low-pass filter, which removes most of the perturbation - which appears to rely mostly on high frequency components. Doing so however also degrades the original recognition performance significantly.

In Table 1, the authors present the token and word error rates for several "filters" (they are not strictly "low-pass", but that is the fundamental idea), and show that "filtered" audio is no longer recognized as the intended adversarial target in most cases, but either as a random word, or the original and correct word, which is still the closest competitor. Unfortunately, the numbers given in Section 5.1's text don't appear anywhere else in the paper (any of the Tables), and I am therefore not sure if the explanations given in the text apply to the numbers shown in the Tables. Specifically, I don't understand how the "ratio" shown in Table 1 is defined, but the success of the mitigation methods seems to be dependent on its reduction. Please clarify.

In Sections 4 and 5.2, when defining "temporal consistency", the authors break each speech utterance into two parts, apply the perturbation only to one half, and compute errors separately for each half. They claim that this is a useful way to identify if the audio has been perturbed or not. There are two problems with this approach:
- speech recognition is a sequence classification problem, and the two halves are not independent of each other (and the authors acknowledge this to some extent). Certainly once a language model has been applied, the output on second half will depend critically on the recognition results on the first half, renedring this approach ineffective
- it seems trivial to identifiy adversrial examples when the ground truth is known

Ultimately, the available space may simply not be sufficient to explain the ideas in enough detail, but the authros need to be much more precise and use standard termnology rather than vague statements such as "expected to be transcribed more differently". There are many such examples (what is the "longest common prefix ratio", how does "signal recovery" work, what does "expected to be transcribed more differently" mean, etc.)

---

### Decision · Program_Chairs · 2018-03-20
**ICLR 2018 Workshop Acceptance Decision**

**Decision:**

Reject

**Comment:**

Based on the reviews, this paper has not been accepted for presentation at the ICLR workshop. However, the conversation and updates can continue to appear here on OpenReview.